# Mechanically Strong and Tailorable Polyimide Aerogels Prepared with Novel Silicone Polymer Crosslinkers

**DOI:** 10.3390/gels8010057

**Published:** 2022-01-12

**Authors:** Zhongxin Zhang, Yurui Deng, Zhiyi Lun, Xiao Zhang, Mingyuan Yan, Pan He, Caihong Li, Yuelei Pan

**Affiliations:** 1State Key Laboratory of Fire Science, University of Science and Technology of China, Hefei 230027, China; zhxinzh@mail.ustc.edu.cn (Z.Z.); dyr@mail.ustc.edu.cn (Y.D.); lunzhiyi@mail.ustc.edu.cn (Z.L.); xiao890829@126.com (X.Z.); yanmyuan@mail.ustc.edu.cn (M.Y.); hppretty@mail.ustc.edu.cn (P.H.); licaihong@mail.ustc.edu.cn (C.L.); 2Institute of Science and Technology Innovation, Civil Aviation University of China, Tianjin 300300, China

**Keywords:** aerogel, polyimide, low-cost cross-linkers, silicone polymers

## Abstract

Polyimide (PI) aerogels were prepared using self-designed silicone polymer cross-linkers with multi-amino from low-cost silane coupling agents to replace conventional small-molecule cross-linkers. The long-chain structure of silicone polymers provides more crosslinking points than small-molecule cross-linkers, thus improving the mechanical properties of polyimide. To investigate the effects of amino content and degree of polymerization on the properties of silicone polymers, the different silicone polymers and their cross-linked PI aerogels were prepared. The obtained PI aerogels exhibit densities as low as 0.106 g/cm^3^ and specific surface areas as high as 314 m^2^/g, and the maximum Young’s modulus of aerogel is up to 20.9 MPa when using (T-20) as cross-linkers. The cross-linkers were an alternative to expensive small molecule cross-linkers, which can improve the mechanical properties and reduce the cost of PI aerogels.

## 1. Introduction

Aerogels are a class of nano-porous solid materials with high porosity, low density, high surface area, and low thermal conductivity [1,2,3]. Due to their unique properties, aerogels have been used in adsorption [4,5], catalysts, supercapacitors [6,7], sensors [8], energy storage [9], and so forth. Over the past few decades, aerogels made from polymer have been developed, such as phenol-formaldehyde resin [10], polyurethanes [11], polyureas [12], polyamides [13], and polyimide [14]. Compared to other aerogels, PI aerogels attracted more attention from researchers due to perfect chemical stability [15], good mechanical properties [16], perfect dielectric performance [17,18] and high decomposition temperatures [19,20].

Typically, PI aerogel is divided into two types: cross-linked PI aerogel and linear PI aerogel. The linear PI aerogels are formed by mixing diamines and dianhydrides in equal proportions to form polyamide acids, which are then formed by thermal or chemical imidization. Due to the existence of large conjugated groups with a planar structure on the polymer chain, the PI aerogel is likely to shrink due to intermolecular interactions [21]. To reduce the shrinkage of samples, in the last decade, PI aerogels were mainly cross-linked by multi-amino small molecules, such as 1,3,5-triaminophenoxybenzene (TAB) [19,22] and (aminophenoxy)silsesquioxane (OAPS) [23] 2,4,6-tris(4-aminophenyl)pyridine (TAPP) and 1,3,5-tris(aminophenyl)benzene (TAPB). However, the high cost and poor flexible properties of PI aerogels by using the above cross-linkers limits the application and promotion of PI aerogels. To solve this problem, the silane coupling agents with amino groups are candidates for PI cross-linkers, such as 3-aminopropyltriethoxysilane (APTES) [24] and bis(trimethoxysilylpropyl)amine (BTSMA) [25]. These cross-linkers ultimately cap the polyimide oligomer through the amide bond, which may lead to a reduction in the thermal stability and aging resistance of the polyimide aerogel due to the weaker stability of the amide bond compared to the imide [26].

Silicone polymers are a class of organosilicon polymers with repeating Si-O bonds as the main chain and organic groups directly attached to the silicon atoms [27]. The Si-O bond on the main chain is highly stable [28], while the organic groups in the side groups provide good compatibility of the Silicone polymers with organic polymers [29,30]. The silicone polymers with multi-amino show good dispersion and excellent compatibility with polyimides in polar solvents [31]. Furthermore, silicone polymers exhibit desirable properties such as excellent heat resistance [30,32]. To the best of our knowledge, these materials are mainly used in the electronics and electrical industries, in coatings, rubber release agents, and hair and skin care products, but there are few reports of their use as aerogel crosslinkers.

In this work, we have synthesized the self-prepared silicone polymers cross-linkers with multi-amino from low-cost silane coupling agents to replace conventional small-molecule cross-linkers. The effect on polyimide aerogels was investigated by controlling the density of the amino group and the degree of branching of the silicone polymers. The cost of the prepared silicone polymer crosslinkers has been reduced by 80% and has a superior modulus of 20.9 MPa, which has promising applications in the fields of thermal insulation materials, battery separators and aerospace.

## 2. Results and Discussion

The silicone polymer cross-linkers with multi-amino were prepared in NMP by controlling the proportion of water according to Figure 1a. To investigate the effect of silicone polymer precursors and the degree of polymerization on the properties of aerogels on silicone polymers, the different cross-linkers and their cross-linked PI aerogels were prepared and the number repeating units of polyimide oligomers is 30. To distinguish between different aerogels, we will define the aerogels in this work in terms of cross-linker synthesis raw material—cross-linker polymerization degree—diamine monomer. As an example, the DT-20-DMBZ means that the polyimide aerogel with DMBZ as diamine crosslinked by a crosslinker with a crosslinking degree of 20 synthesized 1:1 by 3-aminopropylmethyldimethoxysilane and 3-aminopropyltrimethoxysilane. The properties of the different polyamide aerogels investigated in this study are shown in Table 1. The density of aerogels shows a slight variation between 0.1–0.11 g/cm^3^, which indicates that the cross-linkers have little effect on the density of aerogels due to their minimal amount. he shrinkage of the aerogel shows a skewed inverse s-shaped curve as the cross-linking agent itself increases its cross-linking point (Appendix A). The shrinkage of the aerogel increases slightly with the degree of polysiloxane polymerization due to the traction of the crosslinker (Appendix A).

### 2.1. Chemical Structure of the PI Aerogels

Fourier transform infrared (FT-IR) spectra of the samples shows absorption peaks at 1370 cm^−1^, 1721 cm^−1^, and 1775 cm^−1^ for the C–N stretching band, C=O asymmetric stretching band, and C=O symmetric stretching band of imide, respectively (Figure 2a,b). As no peak is noticed for the anhydride stretching band at 1860 cm^−1^, it is assumed that anhydride has been consumed. Peaks disappear at 1660 cm^−1^ and 1535 cm^−1^ for the amide C=O stretching band and amide C–N respectively, indicating that the imidization reaction is complete. The bands at 1807 cm^−1^ and 980 cm^−1^ are absent, ruling out the presence of polyimide isomers. The bands at 720 cm^−1^, 1440 cm^−1^, 2858 cm^−1^, and 2925 cm^−1^ for the C–H wagging band, out of plane bending band, symmetric stretching band and asymmetric stretching band, respectively, are contributed to the aliphatic chains of cross-linkers.

Figure 2c shows solid NMR spectra of selected formulations from this study. All spectra contain an imide carbonyl peak at 166 ppm as well as broad aromatic peaks between 108 and 140 ppm, characteristic of these polyimides. Aerogels made with DMBZ show a peak at 153 ppm characteristic of the aromatic ether carbon. Aerogels made with DMBZ give an aliphatic peak at 19 ppm for the pendant methyl groups. The peak of the cross-linkers is absent due to the low content of crosslinkers in aerogels.

### 2.2. The Porous Structure of the PI Aerogels

The isotherms of aerogels cross-linked with all samples exhibit IUPAC type IV curves with an H1 hysteresis loop according to Figure 3a,d, indicating that the aerogels consist predominately of three-dimensional continuous mesopores. The specific surface area of the aerogels was relatively uniformly distributed, all in the range of 300–500 m^2^/g, and the maximum specific surface area is 469 m^2^/g. It reflects the excellent homogeneity of the silicone polymer cross-linked polyimide aerogels. Under the same conditions, increasing the degree of branching of the crosslinker decreases the specific surface area of the aerogel (Figure 3b). This is mainly because the increase in branching leads to an increase in the cross-linking of the aerogel on the silicone polymer chains, which is not conducive to the self-adjustment of the aerogel’s structure during aging. At the same degree of branching, the increase in the polymerization degree of silicone polymers leads to a slight decrease in specific surface area (Figure 3e). This effect becomes negligible as the degree of branching increases. According to the pore volume versus pore size plots of aerogels (Figure 3c,f), the pore size distribution of aerogels was basically in the range of 2–100 nm. As the branching of the silicone polymers increases, the pore size distribution becomes progressively narrower and the pore size becomes progressively smaller, with a minimum of 11 nm. This is probably because the degree of branched silicone polymers increases the crosslinking of the aerogel and enhances intermolecular chain interactions. It leads to greater resistance to the self-assembly of the polyimide chains during the stacking process, resulting in a more uniformly distributed structure. The shrinkage of aerogels cross-linked with different silicone polymers supports this conclusion. The shrinkage of aerogels decreases as the degree of silicone polymers branching increases, while the polymerization degree of siloxanes has no significant effect on the shrinkage of aerogels.

### 2.3. The Morphology of the PI Aerogels

Scanning electron microscopy (SEM) results of representative aerogels in the study are shown in Figure 4a,b. The morphology of the samples prepared with the different cross-linking agents was very similar to that of aerogels prepared with other small molecule cross-linking agents [19,21]. It indicates that the cross-linkers show good compatibility and dispersion with the polyamide acid oligomers in NMP solutions, which contributes to the stability and aging resistance of aerogels. EDX elemental mappings also show perfect dispersion of silicone polymers and polyimides in solution due to Figure 4c–f. All aerogels are made up of smooth fibers. It is worth noting that the fibers of the aerogel behave more coarsely as the degree of branching of the crosslinker gradually increases. This is likely because the degree of self-crosslinking increases and the entanglement tension of the corresponding aerogel increases, while the degree of branching of the crosslinker molecules increases at a certain degree of polymerization.

### 2.4. The Mechanical Properties of the PI Aerogels

To investigate the effect of crosslinkers on the mechanical strength of aerogels, all samples in the study were compressed. The Young’s modulus of the aerogel is measured at the initial slope of the stress–strain curve, some of which are shown in Figure 5a,b. Similar to aerogels cross-linked with small-molecule cross-linkers, the aerogels with silicone polymers exhibit a linear elastic region up of 0.04 to 0.8 strain. The Young’s modulus is taken as the initial slope of the stress strain curve of the aerogels cross-linked with poly-siloxane. The Young’s modulus is in the region of 7–21 MPa, with a maximum of 20.9 MPa. It is worth noting that the Young’s modulus of the aerogel decreases as the amino density increases, when the silicone polymers crosslinker is a linear structure (Figure 5c). This may be attributed to the fact that, at higher densities of amino, the self-shielding of the alkyl chain affects the dispersion of the crosslinker, resulting in a reduction in its modulus. The Young’s modulus of the aerogel gradually increases with increasing branching of the silicone polymers (Figure 5d). This indicates that the increased crosslinking of the crosslinker also improves the mechanical properties of the aerogel. The branching of the silicone polymers shows almost no effect on the compression modulus of the aerogel due to the high molecular weight of silicone polymers. At this strength, three 0.1 g aerogel samples can support the gravity of a 70 kg adult man (Appendix A). The change in the compression modulus of the aerogel is negligible when the degree of polymerization of the poly-siloxane increases.

The stress–strain curves of PI aerogel films are cross-linked with silicone polymers as shown in Figure 6a,b. In general, the tensile modulus should be similar to the compressive modulus. The lower tensile modulus observed in the tests may be due to the inhomogeneity of the films and the structure of the silicone polymers. Similar to compression, the tensile modulus and tensile stress at break of aerogel films increase considerably with the amount of polysiloxane branching (Figure 6c), while the degree of polymerization of the crosslinker has a negligible effect (Figure 6d). The lower modulus of elasticity and elongation at the break of aerogels cross-linked with polysiloxanes compared to aerogels cross-linked with small molecules may be related to the structure of the silicone polymer. At the same time, aerogels exhibit excellent cuttable properties (Figure 6f), which offer more possibilities for their application and dissemination.

### 2.5. The Thermal Stability of the PI Aerogels

TGA of silicone polymers cross-linked PI aerogels was carried out from room temperature to 800 °C in nitrogen. The graphs of PI aerogels’ TGA profiles are shown in Figure 7a. There is a more significant weight loss due to solvent evaporation between the start of heating to 190 °C. Then, a heat-absorbing peak appears on the TGA curve at 220 °C due to the decomposition of methyl. The main chain of polyimide molecular chains with a hypercodone structure begins to break down at 520 °C. According to the TGA curve, the final decomposition of the sample is divided into two stages: the opening ring of the amide and the pynesis of the molecular chain. At the beginning, the imide structure on the main chain begins to break down gradually. All structures on the main chain then begin to degrade gradually and carbonize in N_2_ gas atmosphere. Discounting the effect of solvent volatilization, all the formulations charred at over 73% in nitrogen. These results are in general agreement with the thermal studies of non-porous, bulk polyimides [31].

Figure 7b shows the comparison of Young’s modulus vs. cost of silicone polymers crosslinkers. The costs of other reagents were obtained by reference to the average prices of Aladdin Reagents, China and Maclean’s Reagents, China, while the cost price of crosslinker in this paper is based on the average price of the reagents consumed by Aladdin Reagent Company, China and Maclean’s Reagent Company, China. The aerogels cross-linked with silicone polymers exhibit more excellent mechanical strength and modulus than aerogels cross-linked with small molecules [23,31]; additionally, the cost of the cross-linker is substantially reduced. Thus, silicone polymers with amino can be a low-cost alternative to cross-linkers for polyimide aerogels.

## 3. Conclusions

PI aerogels were prepared with self-designed silicone polymers capped polyamide oligomers. The effect on polyimide aerogels was investigated by controlling the density of the amino group and the degree of branching of the silicone polymers. Aerogels cross-linked with T-50 have the highest mechanical strength and tensile modulus. These were formulated aerogels with the smallest specific surface area and pore size, probably due to their lower compatibility with polyimide (faster phase separation), resulting in higher density and lower porosity. The aerogels cross-linked with silicone polymers exhibit superior mechanical strength and modulus than the aerogels cross-linked with small molecules; additionally, the cost of the cross-linker is substantially reduced. Thus, silicone polymers with amino can be a low-cost alternative to cross-linkers for polyimide aerogels.

## 4. Materials and Methods

### 4.1. Materials

BPDA, DMBZ, dimethydimethoxysilane, 3-aminopropymethyl-dimethoxysilane, and 3-aminopropyltrimethoxysilane were purchased from Aladdin Chemical Reagent Co. (Aladdin, Shanghai, China). N-methyl-2-pyrrolidinone (NMP), anhydrous acetic anhydride, acetone, and pyridine were purchased from Sinopharm Chemical Reagent Co. Ltd. (Sinopharm Chemical, Shanghai, China). All reagents were used without further purification.

### 4.2. Preparation of Silicone Polymers

Different silane coupling agent molecules dissolved in the polar solvent NMP, and then the appropriate amount of water was added in proportion to obtain silicone polymer cross-linker solutions with different amino densities and branching degrees. The final cross-linker solution was formulated to 0.2 M (amino concentration). The silicone polymers are named by their monomer and degree of polymerization, such as D_0_D-50, meaning that the cross-linker is a silicone polymer with a degree of polymerization of 50 prepared in a 1:1 ratio of dimethyldimethoxysilicane and 3-aminopropylmethyldimethoxysilane. The preparation of the D_0_D-20 cross-linkers is an example. Dimethyldimethoxysilane (120 mg, 1 mmol) and 3-aminopropyldimethoxymethylsilane (163 mg, 1 mmol) dissolved in 50 mL NMP, then water (34 mg, 3.8 mmol) was added. The mixture was stirred under sealed conditions for 12 h at room temperature to form the D_0_D-20 cross-linkers.

### 4.3. Preparation of PI Aerogels

Figure 1b shows the reaction of preparing process polyimide aerogels. Polyamide acid oligomer with the repeat unit n of 30 obtained by using the ratio of dianhydride BPDA (n + 1) to diamine DMBZ (n) was synthesized in NMP. As an example, the preparation of the 10% PI aerogels with PSMA cross-linked and n = 30. DMBZ (9.3 mmol, 1.98 g) was dissolved in 55 mL of NMP under magnetic stirring. Then BPDA (8.4 mmol, 2.648 g) was added. The mixture was stirred for 30 min at room temperature to form the oligomer. Subsequently, the solution of silicone polymers was added with stirring until a homogeneous solution was obtained. Then acetic anhydride (74.4 mmol, 7.26 mL, 8:1 M ratio to DMBZ) and pyridine (74.4 mmol, 7.26 mL, 1:1 M ratio to acetic anhydride) were added into the solution. Afterwards, the solution was poured into the polypropylene molds and gelled. Then, the gels were aged for 24 h in the mold at room temperature. Next, the wet gels were sequentially immersed in 75%, 25% NMP solution in acetone, and 100% acetone for 24 h for solvent exchange. Finally, the monolithic PI aerogels were obtained using supercritical fluid CO_2_ drying (55 °C, 12 MPa, 96 h), followed by vacuum drying for 12 h at 60 °C to remove any residual acetone.

### 4.4. Characterization

The bulk density, $\rho_{b}$, was calculated by dividing the weight of the sample by the volume. The shrinkage was calculated from the volume difference between the gel sample and the polypropylene mold. The porosity (Π) was calculated based on the bulk and skeletal densities as shown in Equation (1):(1)$$\Pi =(1-\frac{\rho_{b}}{\rho_{s}})\times 100\%$$

The specific surface areas and total pore volume of pores were measured by nitrogen sorption isotherms with standard Brunauer–Emmett–Teller (BET) analysis (Tristar II 3020M, Micromeritics Instrument Corporation, Norcross, GA, USA). The pore size distributions (PSD) were estimated by the Barrett–Joyner–Halenda (BJH) method (Tristar II 3020M, Micromeritics Instrument Corporation, Norcross, GA, USA). All samples were degassed at 80 °C for 24 h under vacuum before analysis. Fourier transform infrared spectra (FT-IR, Nicolet 8700, Thermo Fisher Scientific, Thermo Fisher Scientific, Waltham, MA, USA) were obtained to investigate the chemical bonding state of these aerogel samples. Solid ^13^C NMR spectra of the polymers were obtained on a Bruker AVANCE AV III 400WB (Bruker, Karlsruhe, Germany) spectrometer using the 89 mm solids probe with magic angle spinning at 15 kHz and cross-polarization. Spectra were externally referenced to the carbonyl peak of glycine (176.1 ppm relative to TMS). A Hitachi SU-8220 (Hitachi, Tokyo, Japan) field emission microscope was used for the scanning electron microscope (SEM) images after 60 s of sputter coating the specimens with platinization. Thermal gravimetric analyses (TGA, TA INSTRUMENTS, New Castle, PA, USA) were performed using a TA model SDT Q600 instrument. Samples were run at a temperature ramp rate of 10 °C per min from room temperature to 800 °C under nitrogen. A uniaxial compression test was performed using an electronic dynamic and static fatigue testing machine (E3000K8953, Instron, Canton, UK) with a constant loading rate of 1 mm/min. The specimens were cut and polished to make sure that the top and bottom surfaces were smooth and parallel. Samples were conditioned at room temperature for 48 h before testing. The diameter and length of the specimens were measured before testing. The Young’s modulus was taken as the initial linear portion of the slope of the stress–strain curve.

## Figures and Tables

**Figure 1 gels-08-00057-f001:**
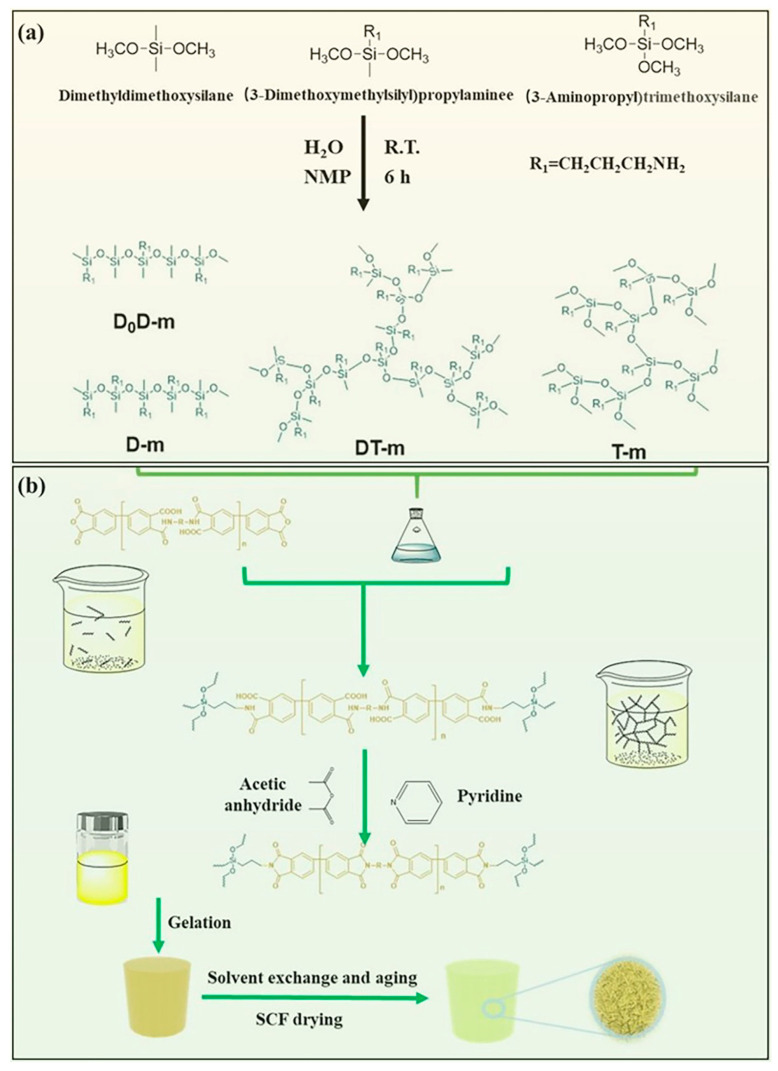
(**a**) Schematic diagram of the crosslinker preparation process; (**b**) Schematic diagram of the polyimide aerogel preparation process.

**Figure 2 gels-08-00057-f002:**
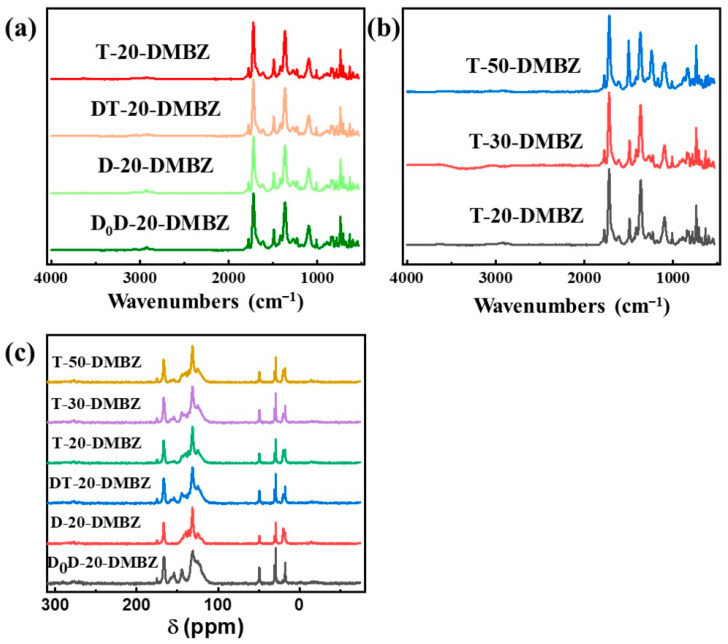
(**a**) Fourier transform infrared (FT-IR) spectra of aerogels cross-linked with silicone polymers prepared from different precursors; (**b**) Fourier transform infrared (FT-IR) spectra of aerogels cross-linked with silicone polymer of different degrees of polymerization; (**c**) ^13^C Solid NMR spectra of representative aerogels in this work.

**Figure 3 gels-08-00057-f003:**
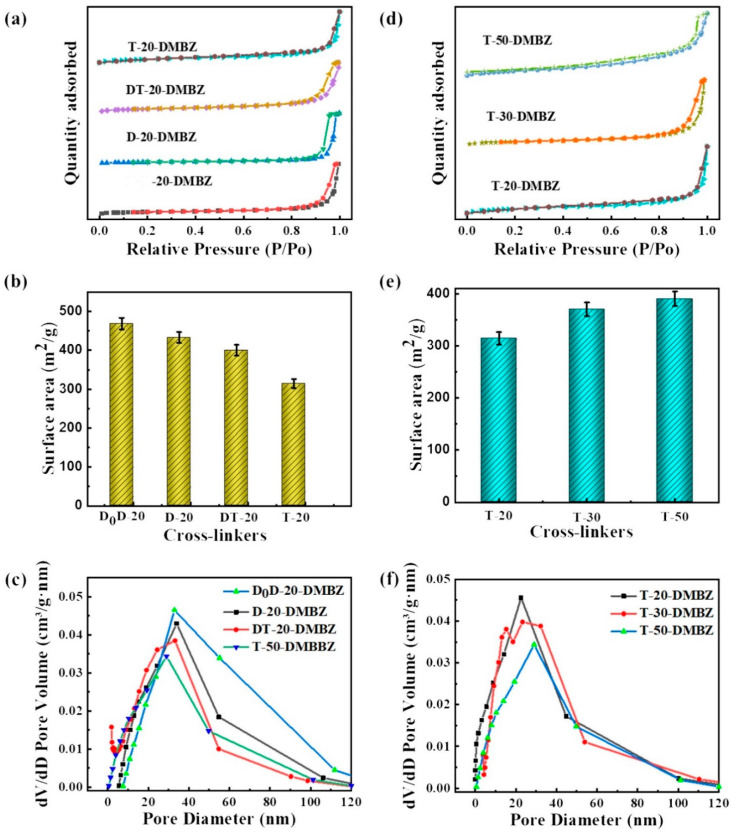
(**a**) N_2_ adsorption-desorption isotherms of polyimide aerogels cross-linked with silicone polymers prepared from different precursors; (**b**) Surface area of aerogels cross-linked with silicone polymers prepared from different precursors; (**c**) Pore volume vs. pore diameter curves for polyimide aerogels cross-linked with silicone polymers prepared from different precursors; (**d**) N_2_ adsorption-desorption isotherms of polyimide aerogels prepared with silicone polymers of polymerization degree; (**e**) Surface area of aerogels prepared with silicone polymers of different degrees of polymerization; (**f**) Pore volume vs. pore diameter curves for polyimide aerogels prepared with silicone polymers of different degrees of polymerization.

**Figure 4 gels-08-00057-f004:**
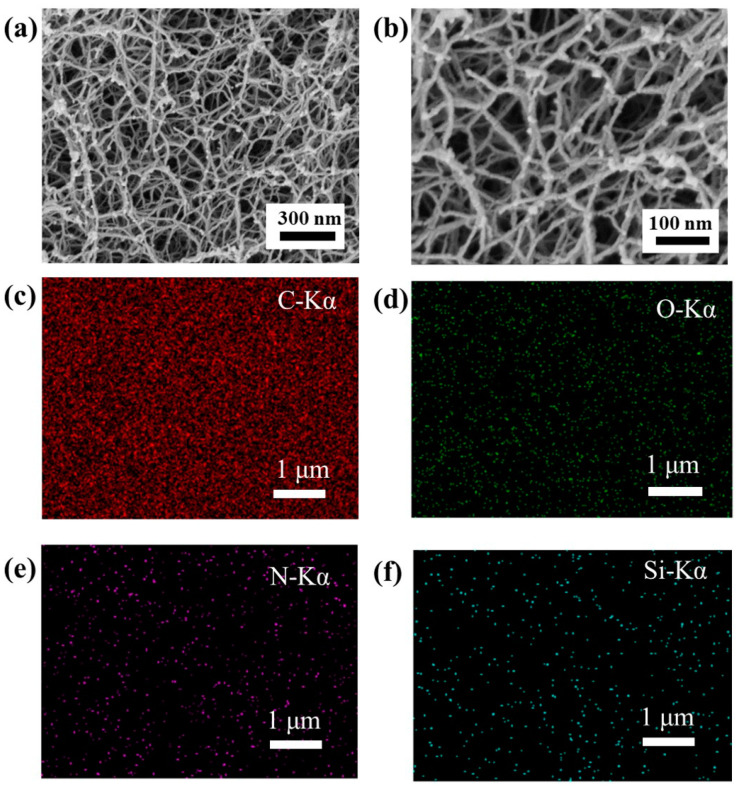
(**a**) SEM image of sample D-20-DMBZ; (**b**) SEM image of sample DT-20-DMBZ; (**c**–**f**) EDX elemental mappings of C (**c**), O (**d**), N (**e**), and Si (**f**).

**Figure 5 gels-08-00057-f005:**
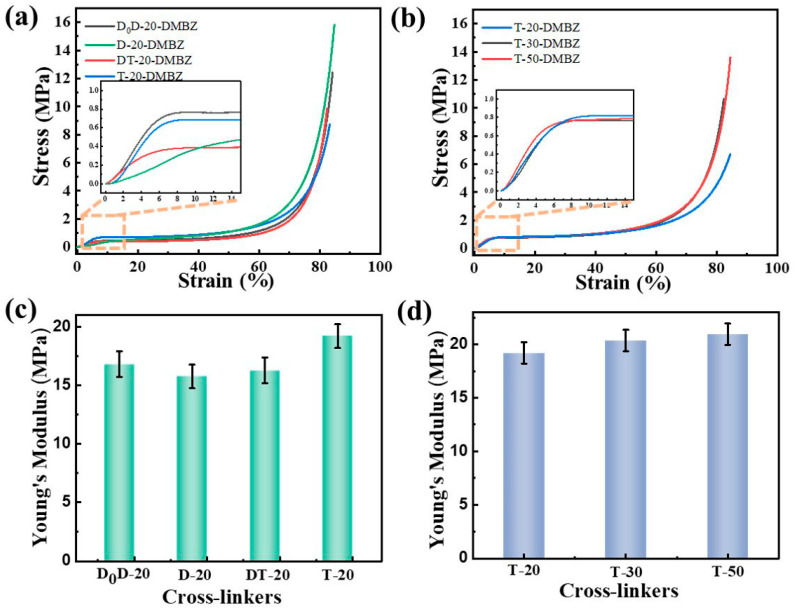
(**a**) Stress–strain curves from compression of polyimide aerogels cross-linked with silicone polymers prepared from different precursors; (**b**) Stress–strain curves from compression of polyimide aerogels prepared with silicone polymers of different degrees of polymerization; (**c**) Young’s modulus of polyimide aerogels cross-linked with silicone polymers prepared from different precursors; (**d**) Young’s modulus of polyimide aerogels cross-linked with silicone polymers prepared with silicone polymers of different degrees of polymerization.

**Figure 6 gels-08-00057-f006:**
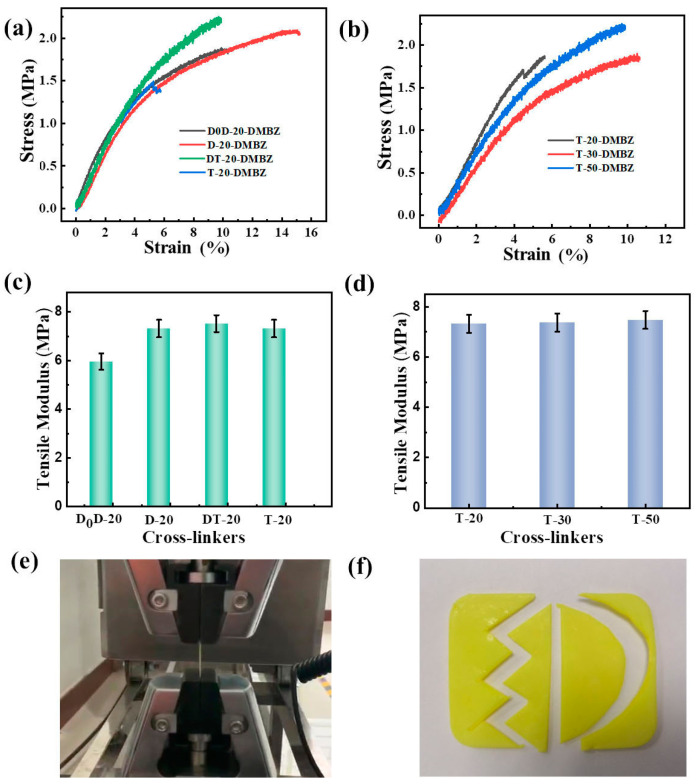
(**a**) Stress–strain curves from stretching of polyimide aerogels cross-linked with silicone polymers prepared from different precursors; (**b**) Stress−strain curves from stretching of polyimide aerogels cross-linked with silicone polymers prepared with silicone polymers of different degrees of polymerization; (**c**) Tensile modulus of polyimide aerogels cross-linked with silicone polymers prepared from different precursors; (**d**) Tensile modulus of polyimide aerogels prepared with silicone polymers of different degrees of polymerization; (**e**) Polyimide aerogel thin film of D-20-1:1 after folding; (**f**) Demonstration of the cutting properties of aerogels.

**Figure 7 gels-08-00057-f007:**
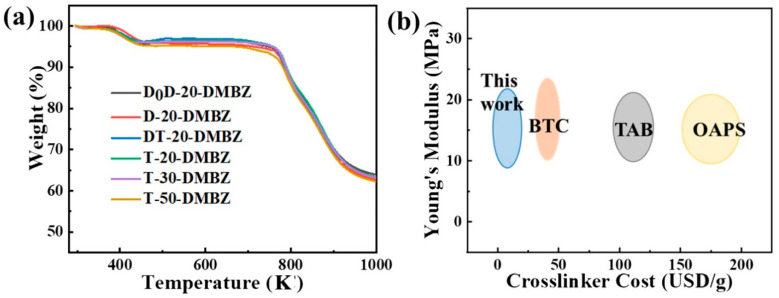
(**a**) Thermogravimetric analysis (TGA) curves in N_2_ of the polyimide aerogels; (**b**) Comparison of Young’s modulus vs. cost of different crosslinkers.

**Table 1 gels-08-00057-t001:** Properties of the different PI aerogels investigated in this work.

Sample	Bulk Density (g/cm^3^)	Shrinkage (%)	Porosity (%)	Surface Area (m^2^/g)	Pore Diameter (nm)
D_0_D-20-DMBZ	0.104 ± 0.003	17.6 ± 1.1	89.6 ± 1.1	469 ± 15	31.4 ± 0.2
D-20-DMBZ	0.101 ± 0.003	18.8 ± 1.1	88.1 ± 1.1	433 ± 14	30.5 ± 0.2
DT-20-DMBZ	0.106 ± 0.002	12.8 ± 1.0	85.5 ± 1.1	400 ± 14	20.1 ± 0.2
T-20-DMBZ	0.106 ± 0.003	14.1 ± 1.0	85.6 ± 1.0	314 ± 12	11.1 ± 0.1
T-30-DMBZ	0.107 ± 0.002	14.4 ± 1.0	87.1 ± 1.0	375 ± 13	12.2 ± 0.1
T-50-DMBZ	0.108 ± 0.002	14.7 ± 1.0	87.4 ± 1.0	390 ± 14	12.5 ± 0.1

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
