# Peer review of "Mechanically Strong and Tailorable Polyimide Aerogels Prepared with Novel Silicone Polymer Crosslinkers"

_gels, 2022, doi:10.3390/gels8010057_

Round 1

Reviewer 1 Report

The study on the prepared polyimide aerogels from novel silicone polymer cross-linkers is interesting both from a scientific and practical point of view. Paper can be recommended for publication. However, the manuscript needs to be reworked further:

  1. In the Introduction, in the 3rd paragraph, the sentence was repeated „Silicone polymers are a class of organosilicon polymers with repeating Si-O bonds as the main chain and organic groups directly attached to the silicon atoms”.
  2. „Figure S1”, „Figure S2” and „Figure S3” appear in the text. Figures S1, S2 and S3 are missing from the publication.
  3. In section „2.5. The thermal stability of the PI aerogels”, under the figure there should be a caption "Figure 7. (a) Thermogravimetric analysis (TGA) curves in N2 of the polyimide aerogels; (b) Comparison of Young's modulus vs cost of different crosslinkers.
  4. In section „2.5. The thermal stability of the PI aerogels”, In the text, instead of „DSC curve”, it should be „TGA curve”. The DSC curve is the result of a Differential Scanning Calorimetry measurement.
  5. Conclusions should be after the „Materials and Methods” section.
  6. In References, there should be an abbreviated journal name.

Author Response

Dear reviewer,

Thanks for your constructive comments which are very helpful to improve the manuscript. According to your comments, we have rewritten and revised the corresponding parts. Now all of them have been listed in the attachment..

Reviewer 2 Report

The authors prepared polyimide aerogels cross-linked with silicone polymers. The aerogels exhibited superior mechanical strength and modulus than those cross-linked with small molecules. Although the manuscript contains interesting results, this reviewer believes that the following points need to be significantly improved before the manuscript can be published in this journal.

1.  It would be better to clearly define the abbreviations of the synthesized aerogels and the meanings of the numbers associated with the abbreviations, and then present the results and discussion.

2.  The author indicated in the abstract that "The long-chain structure of silicone polymers provides more crosslinking points than small-molecule crosslinkers, thus improving the mechanical properties of polyimide." What data shows that long-chain crosslinkers have more crosslinking points than small molecular crosslinkers in aerogels? (Are there more unreacted sites that are not cross-linked in the small molecular crosslinkers?)

3.  The number of pixels in Fig. 1 is low and the structure of the compound is not clear, so it needs to be changed to a clearer figure. The caption of Fig.1 is unintelligible.

4.  There is an error in the axis information of the data showing the stress-strain curves. The words stress and strain are shown reversed in the x-axis and y-axis. This is quite a fatal error.

5.  It looks like all the materials are fracturing at the same value of strain, is this normalizing? Or is it really breaking at the same strain? If this is the case, then the physical properties of the cross-linked point will not change significantly whether it is a polymer or a small molecule.

6.  There are two Fig. 6, so the latter Fig. 6 is probably an error of Fig. 7.

7.  If polymer cross-linking can improve the physical properties of aerogels, it is expected that the longer the molecular chain length between the cross-linked points, the larger the volume change. It would be necessary to compare the temperature-responsive rate of volume change for all samples.

8.  Repeated tensile tests should be performed to see if hysteresis based on the interaction between the polymers between the crosslinking points can be observed.

Author Response

(The authors gave the same response as above.)

Round 2

Reviewer 2 Report

Since the strain values in the answer to question 8 differed by one order of magnitude from the data (Fig. 5(a)) in the manuscript, this reviewer had serious doubts about the data of the materials described in the manuscript. Therefore, this reviewer judged that the manuscript should not be published in this journal unless the reliability of the data in the manuscript is guaranteed.

Author Response

Dear Reviewer,

Thanks for your constructive comments which are very helpful to improve the manuscript. We would like to further explain about your doubts about the authenticity of the data in our manuscript. The reason for your question about the data is that you found a significant difference between the fracture strain we provided in our response comments (around 7%) and the fracture strain in the manuscript (around 80%). However, two sets of data on mechanical properties exist in the manuscript: tensile and compression stress-strain curves, and they are different. As you mentioned, the compressive fracture strain of the aerogel is basically around 80%, which can be reflected in Figures 5a and 5b in the manuscript. However, the tensile strain at break of the aerogel was basically in the range of 6%-15% (Figure 6a and Figure 6b). The strain at break of the sample T-20-DMBZ selected in our response to comments was 6%, which is consistent with the data (around 7%)used in the manuscript. Furthermore, the compressive stress-strain curve shows that the elastic compression range of almost all samples is within 8%. It is consistent with the tensile fracture strain. Therefore, you may have misinterpreted the compression strain in the manuscript (Figure 5) as a tensile strain (Figure 6), which leads to the illusion of seemingly mismatched data.

Round 3

Reviewer 2 Report

This reviewer would like to thank the authors for their prompt response to questions about the data from this reviewer. Since the reliability and reproducibility of the data in this manuscript have been confirmed and the questions raised by this reviewer have been resolved, this reviewer considers this paper to be ready for publication in this journal.

Author Response

Dear Reviewer:

       Thanks vevr much for your kind work and consideration on publication of our paper. On behalf of my co-authors, we would like to express our great appreciation to you.